# Passive Grouping Enhances Proto-Arithmetic Calculation for Leftward Correct Responses

**Maria Loconsole *** , **Lucia Regolin** and **Rosa Rugani**

Department of General Psychology, University of Padua, 35131 Padua, Italy
* Correspondence: maria.loconsole@unipd.it

**Abstract:** Baby chicks and other animals including human infants master simple arithmetic. They discriminate 2 vs. 3 (1 + 1 vs. 1 + 1 + 1) but fail with 3 vs. 4 (1 + 1 + 1 vs. 1 + 1 + 1 + 1). Performance is restored when elements are grouped as 2 + 1 vs. 2 + 2. Here, we address whether grouping could lead to asymmetric response bias. We recoded behavioural data from a previous study, in which separate groups of four-day-old domestic chicks underwent an arithmetic task: when the objects were presented one-by-one (1 + 1 + 1 vs. 1 + 1 + 1 + 1), chicks failed in locating the larger group irrespective of its position and did not show any side bias; Experiment 1. When the objects were presented as grouped (2 + 1 vs. 2 + 2), chicks succeeded, performing better when the larger set was on their left; Experiment 2. A similar leftward bias was also observed with harder discriminations (4 vs. 5: 3 + 1 vs. 3 + 2), with baby chicks succeeding in the task only when the larger set was on the left (Experiments 3 and 4). A previous study showed a rightward bias, with tasks enhancing individual processing. Despite a similar effect in boosting proto-arithmetic calculations, individual processing (eliciting a right bias) and grouping (eliciting a left bias) seem to depend on distinct cognitive mechanisms.

**Keywords:** numerical discrimination; proto-arithmetic; hemispheric specialization; spatial bias; domestic chicken

## 1. Introduction

Evidence of simple arithmetic abilities has been reported in human infants [1–3] and non-human animals, including mammals (e.g., chimpanzees [4,5], monkeys [6,7], elephants [8], dogs [9], rats [10], mice [11]), birds (e.g., parrots [12], jackdaws [13]), fish (e.g., mosquitofish [14], archerfish [15]), reptiles (e.g., tortoises [16], lizards [17]), and even invertebrates (e.g., honeybees [18], crickets [19], spiders [20]). Being able to process and respond to simple numerical information confers a strong ecological advantage in several contexts, e.g., territorial fighting [21], food-hoarding [22], and defence from predation [23]. Young domestic chickens have been widely investigated for their numerical capabilities [24], being a golden model for studying the early emergence of cognitive traits (they are extremely precocial in motor and perceptual development) and controlling the role of experience (they can be tested in the very early stages of life and can be hatched in a controlled environment). Previous studies on precocial species (included the domestic chicken) showed that early exposure to both natural or artificial objects prompts perceptual learning [25–27]. If baby chicks are separated from such objects, they actively search for them and are spontaneously motivated to stay by the larger set [25,28]. This allows for the study of spontaneous numerical abilities in the absence of numerical training. In addition, the absence of a corpus callosum in birds and the nearly complete decussation of their optic fibres at the optic chiasm allow for the study of the neuroanatomical basis of cognitive abilities without resorting to invasive techniques [29]. Being that the two hemispheres are virtually isolated, it is possible to infer hemispheric dominance from the behavioural responses: those under the control of one hemisphere will be preferentially expressed in the

contralateral space [30]. Recent studies have exploited this characteristic feature of chicks' neuroanatomy to investigate spatial biases, hence the involvement of the two hemispheres, in proto-arithmetical tasks. A first study showed that when chicks were tested with the 5 vs. 10 and 6 vs. 9 comparisons, in which objects were individually (i.e., one-by-one) presented and hidden behind two opaque panels, they were better at locating the larger group when it was on their right [31]. This was attributed to a tendency to associate spatial and numerical responses, having the small numerosities on the left and the large numerosities on the right side [32,33]. In a second work, we deepened these results by showing that the same right bias was present even when chicks faced a very complex numerical discrimination, such as 3 vs. 4, following a similar one-by-one presentation. Yet this facilitation emerged only when chicks were supported in the discrimination by an auxiliary cognitive strategy, namely individual recognition, either during rearing or testing did each element depict individual face-like features that made it unique (for a detailed explanation see [25]). When the chicks were reared and tested with all identical stimuli, with the same face-like features depicted on each element, they failed in the overall discrimination and showed no spatial bias [34]. We argued that when the task is too complex and the performance is not supported by any additional strategy, the chicks cannot distinguish between the two sets; hence, they do not process the stimuli in the first place and choose randomly. Vice versa, when they can rely on a cognitive strategy enabling discrimination, they present a rightward bias. This is in line with previous literature [31] and with the idea of a spontaneous tendency to associate the left and right space with small and large numerosities, respectively. In addition, a right bias could suggest higher left hemispheric activation. This could be due to the fact that the employed stimuli have a positive social valence for the chicks, as chicks usually develop a strong attachment to their imprinting objects [25,28,35] and prefer face-like stimuli over non face-like ones [36,37]. The left hemisphere responds to positive valence [38,39] and to the processing of categories, such as conspecific vs. heterospecific [36,37,40].

We wondered whether, in a similar discrimination task that does not support category-based responses or individual recognition, chicks would present the same rightward bias, or rather a different response pattern from recruiting different brain areas. To this aim, we recoded data from a previous study [41] in which 4-day-old chicks were tested with the same 3 vs. 4 discrimination with elements presented as grouped (a strategy that proved effective in supporting performance in the 3 vs. 4 discrimination). It also supported a more complex discrimination (4 vs. 5) when combined with a less demanding testing session, being that this was divided into smaller blocks, spaced out by 2 h breaks.

## 2. Materials and Methods

For a detailed explanation of the employed methodology, we refer the reader to the original paper [41].

### 2.1. Subjects and Rearing Conditions

Forty-three female domestic chicks (*Gallus gallus*) participated in the original study. Newborn chicks were obtained weekly from a local commercial hatchery (Agricola Berica, Montegalda, Vicenza, Italy). The chicks were housed individually in standard metal cages (28 × 32 × 40 cm) at controlled temperature (28°C–31 °C) and humidity (68%). Food and water were available ad libitum. The cages were constantly lit (24 h/day) by fluorescent lamps (36 W), located 45 cm above the floor. Within its cage, each chick was reared with a set of seven (in Experiments 1 and 2) or nine (in Experiments 3 and 4) identical bidimensional red plastic squares (2.5 × 2.5 cm), hanging in the cage via a thin thread. There is wide literature suggesting that chicks that are reared in such a condition develop an attachment toward the rearing stimuli and become highly motivated in rejoining them when separated. In particular, when given the choice between two sets of such stimuli, they will prefer the larger one [28,35,41].

The experiments complied with all applicable national and European laws concerning the use of animals in research and were approved by the Italian Ministry of Health (per-

mit number: 32662 emitted on 10/1/2012). All procedures employed in the experiments included in this study were examined and approved by the Ethical Committee of the University of Padua (Comitato Etico di Ateneo per la Sperimentazione Animale—C.E.A.S.A.) as well as by the Italian National Institute of Health (N.I.H). At the end of the experimental procedures, the chicks were donated to local farmers.

### 2.2. Training and Test

The chicks were tested on their third day of life. Two hours before the test, they underwent a pretraining session, aimed at acquainting them with the experimental arena that consisted in a white circular space (95 cm Ø, 30 cm outer wall height). Adjacent to the inner wall of the arena, there was a starting box (10 × 20 × 20 cm) closed by a transparent glass partition (20 × 10 cm), so that while the chick was kept there, it could see the inner arena and the stimuli. A single blue opaque panel was placed in front of the starting box. At the beginning of each trial, the chick could see one of the imprinting stimuli being moved from in front of the starting box to behind the panel. Then, the glass partition was removed, and the chick was free to explore the arena and rejoin the imprinting object behind the panel. The training was over after three consecutive successful trials (the chick rejoined the imprinting object immediately after being released in the arena).

A complete testing session comprised 20 trials for each subject. The test was conducted using the same arena as training, with the only difference that in front of the starting box there were two opaque panels, one on the left and one on the right. The chick was shown two sets of objects disappearing each behind a panel (Figure 1). The time of presentation for each stimulus/set of stimuli was controlled for, averaging the duration of stimuli presentation for each testing trial: approximately 6 s for hiding the stimuli between each panel, with a 3 s break between the disappearance of the first set and the beginning of the second display. A further 3 s elapsed from the disappearance of the second set and the beginning of the trial (i.e., lifting of the transparent partition).

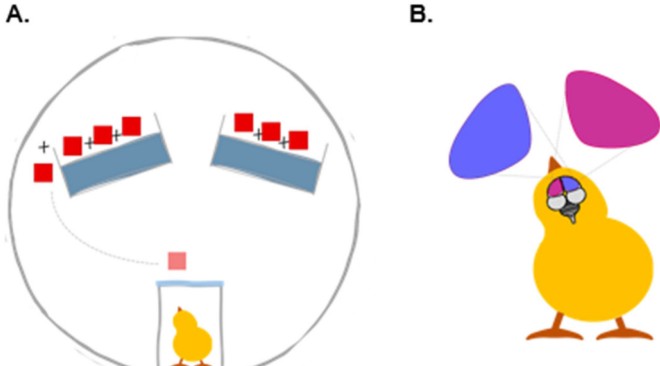

**Figure 1.** (**A**) Scheme depicting the experimental arena and the paradigm employed at test. In this example, the stimuli are displayed as in Exp.1. The chick is restrained in a starting box from where it can see the stimuli being presented in the arena. In half of the trials (i.e., 10 trials), the larger set disappears behind the left panel (as in this example). In the other half, the larger set disappears behind the right panel. (**B**) Simplified representation of the chicks' hemispheric activation in response to lateralized information. Due to a virtually complete decussation of optic nerve fibres at the optic chiasm, each hemisphere mainly processes sensory inputs coming from the contralateral hemispace. As such, a better performance in locating the larger set on the left side could indicate a higher activation of the right hemisphere (in blue), while a better performance in responding to the left side could be indicative of a higher activation of the left hemisphere (in purple).

Once released, the chick could circumnavigate one panel only and rejoin the set behind it. After a few seconds, it was removed from the arena and placed back in the starting box for a new trial. In line with previous studies, the experimental hypothesis was that if chicks could discriminate between the two sets, they would have chosen the one occluding the

larger numerosity [35,41]. The position (left/right) and the size (smaller/larger) of the first set shown to the chick were counterbalanced between trials. In the first three experiments, each chicks underwent all trials in one single session.

In Experiment 1, the chicks were tested with the 3 vs. 4 comparison, following a one-by-one presentation (1 + 1 + 1 vs. 1 + 1 + 1 + 1). In Experiment 2, the chicks were tested with the 3 vs. 4 comparison, presented as grouped (1 + 2 vs. 2 + 2). In Experiment 3, the chicks were tested with the 4 vs. 5 comparison, presented as grouped (1 + 3 vs. 2 + 3). In Experiment 4, the chicks were tested under the same conditions as in Experiment 3 (1 + 3 vs. 2 + 3), the only difference being that the test was divided into 4 sessions of 5 trials each, with a 2 h interval between each session.

Besides the main goal of this study (see [41]), the employed paradigm allows for investigation of possible spatial biases and hemispheric dominance. To solve the task and to locate the larger sets, the chicks needed to: (i) keep track of all objects' displacements (these being presented one-by-one or in small groups); (ii) create an internal representation of each set, also including its spatial position; (iii) locate the larger set by comparing the mental representations of their spatial position. The usage of complex numerical discriminations constitutes an additional challenge for the chicks' cognitive system. To overcome cognitive overload, chicks might rely on hemispheric specializations that could support their performance. If so, this would result in a lateralized behaviour for which chicks perform better when required to respond on a specific side of the arena. A previous study [34] showed that paradigms that support individual processing as a cognitive strategy result in boosting correct responses to the right side. While it is known that grouping is an effective strategy to improve arithmetic performance in chicks (similarly to individual processing), a side bias related to this modality of presentation has not been described thus far.

### 2.3. Data Analysis

Statistical analyses were performed in R 4.2.2 [42]. We set as the dependent variable the binomial choice of the chicks (1 = choice of the larger set; 0 = choice of the smaller set), and as the independent variable the spatial location (left or right) of the larger set. As there were multiple observations for each chick, we employed generalized linear mixed effect models (R package: lme4 [43], with subject IDs as the random effect. Subsequently, we performed a post hoc analysis with Bonferroni correction (R package: emmeans [44]) to determine the direction of the predictor. Graphs were generated using ggplot2 [45].

### 3. Results

In the original work by Rugani and colleagues (2017) [41], chicks failed in the 3 vs. 4 comparison when objects were presented individually (Exp.1). This result was replicated in subsequent studies employing a similar methodology [25,35]. Bird performance was restored when objects were presented in small groups (1 + 2 vs. 2 + 2, Exp.2). When increasing the difficulty of the discrimination, the grouping strategy was ineffective and chicks chose by chance (1 + 3 vs. 2 + 3, Exp.3). However, dividing this test into smaller blocks, spaced out by a 2 h pause, was sufficient for the chicks to solve the task and locate the larger set (Exp.4). For a detailed interpretation of these results, please refer to the original study [41].

We found that in Exp.1 (Figure 2A), there was no effect of the spatial position of the larger set ($X^2 = 1.218$, $p = 0.27$). The chicks failed both when the correct location was on their left (P(larger) = 0.478, SE = 0.063, z = $-0.351$, $p = 0.726$) or on their right (P(larger) = 0.411, SE = 0.061, z = $-1.418$, $p = 0.156$) without differences between the two performances (ratio(left/right) = 1.31, SE = 0.321, z = 1.103, $p = 0.27$). In Exp.2 (Figure 2B), we found an effect of the spatial position ($X^2 = 5.488$, $p = 0.019$), for which the chicks succeeded in locating the larger set both on the left (P(larger) = 0.8, SE = 0.046, z = 4.801, $p < 0.0001$) and on the right (P(larger) = 0.675, SE = 0.059, z = 2.727, $p = 0.006$), but overall performed better on the left side (ratio(left/right) = 1.93, SE = 0.543, z = 2.343, $p = 0.019$). In Exp.3

(Figure 2C), we found an effect of the spatial position ($X^2$ = 4.823, *p* = 0.028). In this case, chicks succeeded in finding the larger set on the left (P(larger) = 0.617, SE = 0.044, z = 2.532, *p* = 0.011), but failed when it was on the right (P(larger) = 0.475, SE = 0.046, z = −0.547, *p* = 0.584), with the two performances being statistically different (ratio(left/right) = 1.75, SE = 0.466, z = 2.196, *p* = 0.028). Similar to Exp.3, in Exp.4 (Figure 2D), there was a significant effect of the spatial position ($X^2$ = 14.002, *p* = 0.0002), with the chicks succeeding on the left (P(larger) = 0.709, SE = 0.041, z = 4.467, *p* < 0.0001) but not on the right (P(larger) = 0.477, SE = 0.045, z = −0.513, *p* = 0.608), and the two performances being statistically different (ratio(left/right) = 2.67, SE = 0.701, z = 3.742, *p* = 0.0002).

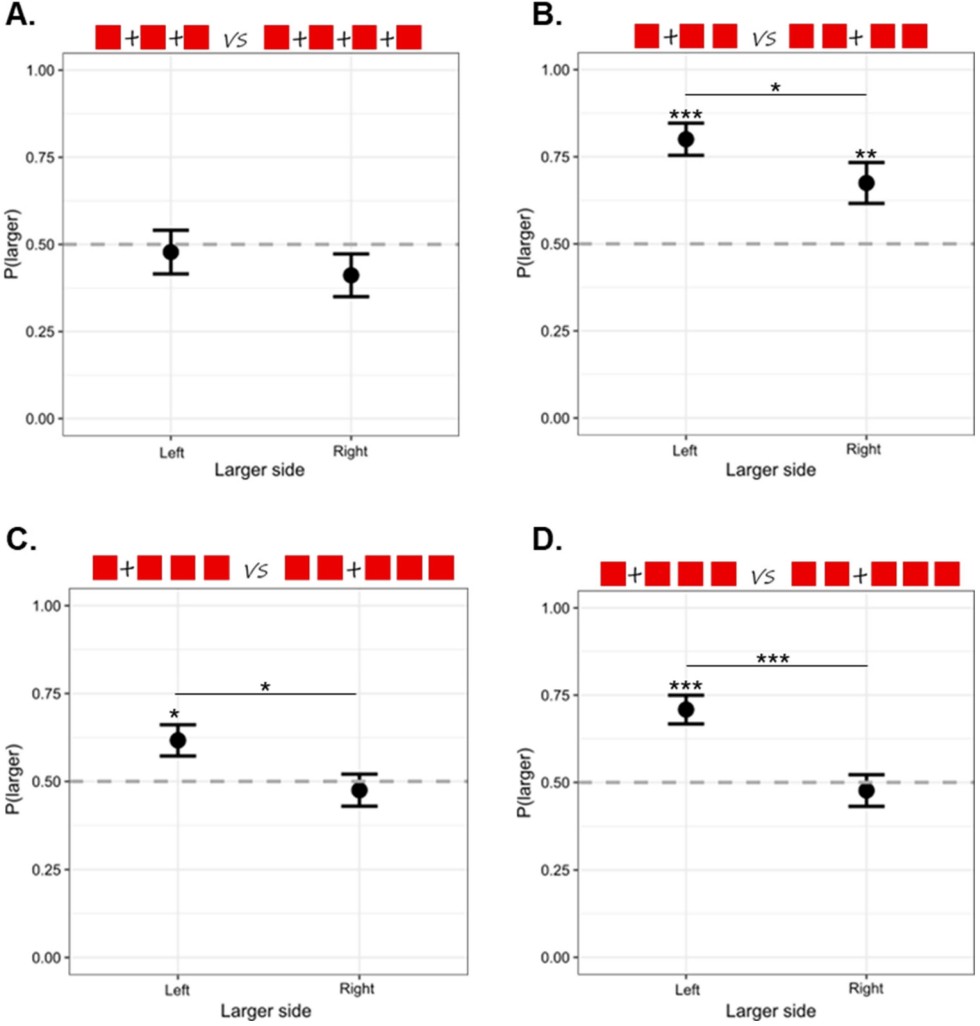

**Figure 2.** On the y axis is the probability of choosing the larger set; on the x axis is the spatial position (left or right) of the larger set. The dashed grey line indicates chance level (*p* = 0.5). The bars indicate the standard error. * *p* < 0.05; ** *p* < 0.01; *** *p* < 0.001 (**A**) Results of Exp.1. Chicks tested with the 3 vs. 4 comparison and elements presented one by one failed in locating the larger set, irrespective of its spatial position. (**B**) Results from Exp.2. Chicks tested with 3 vs. 4 and elements presented as grouped succeeded in locating the larger set both on the left and right, but performed better when the correct response was the left side. (**C**) Results from Exp.3. Chicks tested with 4 vs. 5 and elements presented as grouped succeeded on the left but not on the right side. (**D**) Results from Exp.4. Chicks were tested with the same condition as in Exp.3, but the testing session is split into 4 blocks of 5 trials each, spaced out by 2 h breaks. Birds succeeded on the left but not on the right side.

## 4. Discussion

A previous study showed that baby chicks are better at locating the larger of two sets of objects when this is hidden behind a panel on their right side [31,34]. Here, we assessed

whether the same occurs when chicks are tested on a similar task that, however, favours a different cognitive strategy, i.e., grouping. We hypothesised that a different presentation of the stimuli could lead to the emergence of different biases, each reflecting a specific brain asymmetry resulting from the employed cognitive strategies. This is in line with the literature on numerical cognition in infants that showed how the modality of stimulus presentation could trigger the emergence of one of two possible numerical systems (i.e., the object-file, involved in precise computation of small quantities, or the analogue-magnitude, involved in approximate computation of larger sets) [46].

Similar to our previous results, we found that when chicks are tested with the 3 vs. 4 comparison, following a one-by-one presentation of all identical elements, they failed in discriminating among the two resulting sets and showed no spatial bias. However, in the experimental conditions when the grouping cognitive strategy was induced, we found a facilitation in the opposite direction to that reported in our previous work [34]. Although a one-by-one presentation of individually different objects favours a right bias, grouping strategies seem to trigger a leftward response pattern, probably reflecting a higher activation of the right hemisphere. It is not surprising to find brain asymmetries in cognitive functions, as they have been widely attested in baby chicks and are thought to provide the animals with some benefits, including increasing neural capacity and sparing cognitive resources [47,48]. However, it is yet to be determined how the difference in the orientation of the bias (i.e., toward the left following grouping presentation and toward the right following individual presentation) arises. Both studies (Rugani et al., 2017, employing a grouping strategy [41], and Rugani et al., 2022, employing an individual recognition strategy [25]) are comparable for age of the subjects, type of stimuli (2D objects), training (brief habituation to the arena and to circumnavigating the panel) and testing (20 trials of dichotomous free choice) procedures. The main difference is in the appearance of the stimuli (identical red squares vs. face-like) and in the modality of presentation (grouped vs. one-by-one). We can safely exclude the appearance of the stimuli as a possible cause of the different results, as the same identical red squares were used in a previous study, where the chicks also showed a rightward bias [31]. In this latter case, the chicks did not require any induced cognitive strategy, as the numerical discriminations employed were relatively easy (i.e., 5 vs. 9 and 6 vs. 10), and, importantly, the test followed a one-by-one presentation of each element (the same as Rugani and colleagues, 2022). As such, we believe that a possible explanation for the leftward bias reported in the current work can be attributed to the stimuli presented as grouped. Possibly, animals have a spontaneous tendency to match the left space with smaller numerosities and the right space with larger ones [32], which could explain the right bias reported in the first study by Rugani and colleagues (2014), where no supporting strategies where induced. Yet different cognitive strategies, each leading to a different hemispheric activation, could act on this spontaneous tendency and modify the facilitation effect for locating the larger set. In the case of Rugani and colleagues (2022), individual face-like recognition could have triggered an asymmetric activation in favour of the left hemisphere (hence a facilitation in responding to the right side), as the task required a categorization process of the stimuli [36,37,40]. In the case of Rugani and colleagues (2017), a grouping strategy could have discouraged individual categorization. A left bias could have instead resulted from chicks stressing the spatial component of the task, i.e., right hemisphere preferentially activates in geographical and topographical tasks in both chicks [49,50] and humans [51,52]. Another relevant observation concerns a possible response to novelty, which usually causes a higher right hemispheric activation in baby chicks [40,48]. In both studies, the one-by-one presentation of the objects at test somehow mirrored chicks' training experience, where a single element was shown and hidden behind the panel. The same training procedure was employed by Rugani and colleagues (2017) to habituate the chicks to the arena and to the experimental procedure, without providing them with a numerosity training, because each training trial involved one single object [41]. Thus, at test, chicks saw a grouped set of objects (2 + 1 vs. 2 + 2, or 1 + 3 vs. 2 + 3) for the first time, and this novelty may have resulted in right hemispheric activation and a

consequent left bias [40,48]. This would be consistent with a previous study that showed the emergence of a left spatial bias in young domestic chicks in a reversal learning task that required them to stress novelty detection and visuo-spatial abilities [53], both functions being largely attested to correlate with a higher right hemispheric dominance.

## 5. Conclusions

This work aimed at investigating the presence of a lateralized response in a proto-arithmetic task in young domestic chicks. Previous studies showed that chicks performed better when the larger set had to be located on their right side, probably reflecting a higher left hemispheric activation. Here, we showed that an opposite response pattern can be registered. When the stimuli are presented as grouped, the chicks have an advantage in locating the larger set when this is on their left side. A right hemisphere activation would be consistent with the spatial nature of the task and with a reaction to novelty, since at test, chicks experience grouping for the very first time.

**Author Contributions:** M.L.: Conceptualization, Formal Analysis, Writing—Original Draft; L.R.: Conceptualization, Writing—Review and Editing, Funding Acquisition; R.R.: Conceptualization, Writing—Review and Editing, Funding Acquisition. All authors have read and agreed to the published version of the manuscript.

**Funding:** This work was funded by the European Union's Horizon 2020 research and innovation program under the Marie Sklodowska-Curie grant (795242) to R.R. and by a PRIN 2017 ERC-SH4–A grant (2017PSRHPZ) to L.R. This work was carried out within the scope of the project "use-inspired basic research", for which the Department of General Psychology has been recognized by the Ministry of University and Research as the Department of Excellence for the period 2018–2022.

**Institutional Review Board Statement:** The experiments complied with all applicable national and European laws concerning the use of animals in research and were approved by the Italian Ministry of Health (permit number: 32662 emitted on 10 January 2012). All procedures employed in the experiments included in this study were examined and approved by the Ethical Committee of the University of Padua (Comitato Etico di Ateneo per la Sperimentazione Animale—C.E.A.S.A.) as well as by the Italian National Institute of Health (N.I.H).

**Informed Consent Statement:** Not applicable.

**Data Availability Statement:** Data are available from the corresponding author upon request.

**Conflicts of Interest:** The authors declare no conflict of interest.

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
