# Peer review of "Passive Grouping Enhances Proto-Arithmetic Calculation for Leftward Correct Responses"

_symmetry, doi:10.3390/sym15030719_

Round 1

Reviewer 1 Report

The current paper reanalyzes earlier datasets from chicks to test whether the grouping of items (as opposed to the one-by-one placement of items) influenced the lateral bias. Earlier research of the authors showed SNARC-like effects in chicks (the left side was associated more readily with small numerosity and thus rightward bias with large numerosity). With grouping, chicks had the opposite tendency. I think the results are interesting and worth publishing. I have only minor suggestions.

When authors refer to the arithmetic abilities of non-human animals, they do not seem to refer to rodents and insects. I suggest a more comprehensive reference to the literature.

One important difference between protocols of individual processing and grouping is the temporal correlates of the sequence of events. I think authors should refer to the temporal correlates and how they ensure that subjects do not respond to those instead of numerosities.

It is also possible that in the grouping condition, subjects respond based on the sub-group of objects (those whose set size is not equal; e.g., 2 vs 3 in the case of 2+2 vs 2+3). Can authors also please comment on this possibility?

It seems like the illustration on top of 2B is incorrect since this condition included a 3 vs 4 comparison.

Author Response

The current paper reanalyzes earlier datasets from chicks to test whether the grouping of items (as opposed to the one-by-one placement of items) influenced the lateral bias. Earlier research of the authors showed SNARC-like effects in chicks (the left side was associated more readily with small numerosity and thus rightward bias with large numerosity). With grouping, chicks had the opposite tendency. I think the results are interesting and worth publishing. I have only minor suggestions.

We thank the Reviewer for their comments and suggestions. We have implemented all of them in the revised version of the manuscript.

When authors refer to the arithmetic abilities of non-human animals, they do not seem to refer to rodents and insects. I suggest a more comprehensive reference to the literature.

We have added citations on rodents and invertebrates to give a more exhaustive picture of the current literature. Please see lines: 27 – 30.

One important difference between protocols of individual processing and grouping is the temporal correlates of the sequence of events. I think authors should refer to the temporal correlates and how they ensure that subjects do not respond to those instead of numerosities.

We thank the Reviewer for this interesting consideration. In the original manuscript on grouping strategies (from which we obtained the re-analysed data) this issue is better discussed. The time of presentation for each stimulus / set of stimuli was controlled for, averaging the duration of stimuli presentation for each testing trial: approximately 6sec for hiding the stimuli between each panel, with a 3 second break between the disappearance of the first set and the beginning of the second display. Further 3 seconds elapsed from the disappearance of the second set and the beginning of the trial (i.e., lifting of the transparent partition). We have added this information in the manuscript; please see lines: 119 – 124.

It is also possible that in the grouping condition, subjects respond based on the sub-group of objects (those whose set size is not equal; e.g., 2 vs 3 in the case of 2+2 vs 2+3). Can authors also please comment on this possibility?

We thank the Reviewer for raising this point, which helped us finding an error in the reported information. In fact, the employed grouping for the 4 vs 5 comparison was 3+1 vs 3+2 (not  2+2 vs 3+2). This allowed for excluding the possibility that subjects where using unequal sub-groups as a cue (i.e., both sub-groups were composed of different units). We apologise for the mistake; we have corrected the information in the manuscript.

It seems like the illustration on top of 2B is incorrect since this condition included a 3 vs 4 comparison.

We have corrected the illustration.

Reviewer 2 Report

An interesting work, written with the correct style of a scientific article, however, due to the duty of the reviewer, I would like to draw the authors' attention regarding:

The construction of the sentence seems to be a "storehouse" gathering items of literature, expanding the thought by 2-3 sentences would look much better (26-28) and help to express the thought of the authors.

The structure of the manuscript seems to be not equivalent. The authors describe in detail the course of the experiment and research methods, but the discussion of the obtained results is insufficient. The research results presented by the authors and the collected literature allow for a more comprehensive analysis.

Correct conclusions, supported by previously obtained results and corresponding to the stated purpose of the experiments. After taking into account all corrections, the manuscript is suitable for publication.

Author Response

An interesting work, written with the correct style of a scientific article, however, due to the duty of the reviewer, I would like to draw the authors' attention regarding:

The construction of the sentence seems to be a "storehouse" gathering items of literature, expanding the thought by 2-3 sentences would look much better (26-28) and help to express the thought of the authors.

We have modified the first sentence so that it includes a more extensive picture of species (and clades) tested for numerical cognition, following the suggestion of Reviewer 1 (see lines: 27 – 30). In line with what suggested by the Reviewer, we have also added a second sentence to better explain the relevance of numerical cognition for non-human species, to better frame the reported literature. Please see lines 30 – 32.

The structure of the manuscript seems to be not equivalent. The authors describe in detail the course of the experiment and research methods, but the discussion of the obtained results is insufficient. The research results presented by the authors and the collected literature allow for a more comprehensive analysis.

We thank the Reviewer for giving us the opportunity to expand on the discussion of our results. In line with this suggestion, we have discussed more in depth several points in the introduction and in the discussion. See lines: 27 – 32; 218 – 224; 222 – 237; 271 – 275.

Correct conclusions, supported by previously obtained results and corresponding to the stated purpose of the experiments. After taking into account all corrections, the manuscript is suitable for publication.

We thank the Reviewer for their comments, we have implemented them in the manuscript.